# Evaluating the psychosocial status of BC children and youth during the COVID-19 pandemic: A MyHEARTSMAP cross-sectional study

**Melissa L. Woodward**[1,2]*, **Abrar Hossain**[1], **Alaina Chun**[2], **Cindy Liu**[1,2], **Kaitlyn Kilyk**[2], **Jeffrey N. Bone**[2], **Garth Meckler**[1,2], **Tyler Black**[3], **S. Evelyn Stewart**[2,3,4], **Hasina Samji**[2,5,6], **Skye Barbic**[7], **Quynh Doan**[1,2]

1 Department of Pediatrics, University of British Columbia, Vancouver, British Columbia, Canada, 2 British Columbia Children's Hospital Research Institute, Vancouver, British Columbia, Canada, 3 Department of Psychiatry, University of British Columbia, Vancouver, British Columbia, Canada, 4 British Columbia Mental Health and Substance Use Services Research Institute, Vancouver, British Columbia, Canada, 5 Faculty of Health Sciences, Simon Fraser University, Burnaby, British Columbia, Canada, 6 British Columbia Centre for Disease Control, Provincial Health Services Authority, Vancouver, British Columbia, Canada, 7 Department of Occupational Science & Occupational Therapy, University of British Columbia, Vancouver, British Columbia, Canada

* melissa.woodward@ubc.ca

**Data Availability Statement:** Data cannot be shared publicly because of identifying features in responses which may impact anonymity. Data are available by contacting Melissa Woodward (via

## Abstract

### Background

Understanding the psychosocial status of children and adolescents during the COVID-19 pandemic is vital to the appropriate and adequate allocation of social supports and mental health resources. This study evaluates the burden of mental health concerns and the impact of demographic factors while tracking mental health service recommendations to inform community service needs.

### Methods

MyHEARTSMAP is a digital self-assessment mental health evaluation completed by children and their guardian throughout British Columbia between August 2020 to July 2021. Severity of mental health concerns was evaluated across psychiatric, social, functioning, and youth health domains. Proportional odds modelling evaluated the impact of demographic factors on severity. Recommendations for support services were provided based on the evaluation.

### Results

We recruited 541 families who completed 424 psychosocial assessments on individual children. Some degree of difficulty across the psychiatric, social, or functional domains was reported for more than half of children and adolescents. Older youth and those not attending any formal school or education program were more likely to report greater psychiatric difficulty. Girls experienced greater social concerns, and children attending full-time school at-

melissa.woodward@ubc.ca) for researchers who meet the criteria for access to confidential data and proper REB approval for secondary analyses from the University of British Columbia Children's and Women's Research Ethics Board. Dr. Tibor van Rooij, Director of Research Informatics at the BC Children's Hospital Research Institute, has agreed to be the second point of contact for data inquiries related to this paper. He can be reached at tibor. vanrooij@bcchr.ca. Dr van Rooij's department at BCCHR facilitates the REDCAP system used for the collection and storage of research data meaning that he has full access, but otherwise Dr. van Rooij has no relationship with the study data.

**Funding:** This study was funded through the BC Children's Hospital Foundation with directed donations from generous supporters, including Rio Tinto. The funders had no role in study design, data collection and analysis, decision to publish, or preparation of the manuscript.

**Competing interests:** The authors have declared that no competing interests exist.

home were more likely to identify difficulty within the youth health domain but were not more likely to have psychiatric difficulties. Considerations to access community mental health service were triggered in the majority (74%) of cases.

## Conclusions

Psychosocial concerns are highly prevalent amongst children and adolescents during the COVID-19 pandemic. Based on identified needs of this cohort, additional community health supports are required, particularly for higher risk groups.

## Introduction

Measures to mitigate viral spread of COVID-19 led to closing of schools and workplaces, social isolation, and the disruption of daily routines across Canada. With goals of reducing viral transmission and lowering health service burden, these measures may have altered the pre-pandemic mental health trajectory. Previous viral epidemics have shown increases in depressive symptoms and post-traumatic stress disorder associated with social isolation [1]. Children and adolescents have been significantly impacted by the COVID-19 pandemic and the associated health restrictions [2]. Early studies evaluating the mental health of children during the COVID-19 pandemic indicate elevated measures of depression, anxiety and psychological distress associated with quarantine [3, 4]. Social isolation and psychological stress may have significantly impacted the mental wellness of children and adolescents, affecting youth in community who might not be presenting at the emergency department (ED).

Questions remain about the psychosocial status of young people in the community during the pandemic, the factors associated with greater risk of mental health concerns, and the resources required to support those needs in community. While it is understood that the pandemic has worsened the pre-existing youth mental health crisis, the true extent of that crisis is unclear, understanding both the magnitude of mental health concerns, but also the types of concerns, differentiating social from psychiatric concerns. Prior research has been largely cross-sectional, focused on specific psychiatric disorders and increasing mental health presentations by children and youth to EDs, or online random sampling with studies of low to moderate quality with some failing to report critical methodological details [2, 5, 6]. Many studies fail to assess the impact of schooling method and those that do are of varying quality and estimate various prevalence of mental health concerns [7, 8]. The extent of psychosocial difficulty in young people remains unclear or how to best allocate social services and mental health supports to address any concerns. Using MyHEARTSMAP, a validated self-assessment tool for children and their guardians, this study aims to estimate the frequency of youth psychosocial and health concerns, highlight demographic associated factors, and identify resource needs of youth in British Columbia (BC).

## Methods

### Study recruitment

We recruited families in British Columbia, kids aged 10–17 or parents with kids aged 6–17 years old, between August 2020 and July 2021. Participants who were unable to communicate in English were excluded. Virtual recruitment was conducted through youth and family-oriented organisations, digital networks, and Angus Reid, a private recruitment company, with

care to recruit a sample representative of the geographic distribution of the BC population [S1 Table]. Details of the recruitment procedure have been published [9]. Approval was granted by the University of British Columbia Children's and Women's Health Centre Research Ethics Board.

## MyHEARTSMAP assessment

MyHEARTSMAP is a digital psychosocial health evaluation for youth [S1 and S2 Figs]. MyHEARTSMAP was developed with clinicians and families to evaluate psychosocial concerns in a community setting and has shown good inter-rater reliability and predictability against clinician assessment in children aged 6–17 years presenting to pediatric emergency in western Canada for non-mental health concerns excluding high acuity presentations or requiring resuscitation. This tool has been well validated for self-assessment for universal screening use [10–13]. Psychosocial measures are assessed for severity using a 4-point Likert-type scale from 0 (no concern) to 3 (severe concern) across 10 sections: home, education and activities, alcohol and drugs, relationships and bullying, thoughts and anxiety, safety, sexual health, mood, abuse, and current professional resources. These sections are grouped into four domains based on resources potentially required: psychiatric, social, function, and youth health, with summative severity scores ranging from 0 (none), 1–3 (mild), 4–6 (moderate), and 7 and above (severe) [S3 Fig]. Domain-specific resource recommendations are triggered by any non-zero score within the sections and participants are asked about any established support services and resources for that concern. Sections may trigger multiple recommendation within a domain. These recommendations can include referral to online resources, mental health teams, redirection to a general practitioner, youth protection agencies and services, and youth health services. Details of assessment development and evaluation have been published [11].

## Study design

After providing informed verbal consent over the phone, participants filled out an online survey to collect demographic- and pandemic-related experience information, followed by the MyHEARTSMAP tool. MyHEARTSMAP could be filled out by a guardian, child, or both at the discretion of the participants. Domain-scores in the 'moderate' or 'severe' range automatically triggered an alert for the research nurse on-call, who contacted the youth or guardian depending on which assessment triggered the recommendation to ensure that urgent issues were addressed. Research nurses were pediatric emergency nurses trained in mental health assessment supported by the principal investigator, a pediatric emergency physician.

## Objectives & measures

Our primary objective was to report the frequency of psychosocial issues for children and adolescents in BC during the COVID-19 pandemic. Our secondary objective was to determine associations between severity of psychosocial concerns and participant characteristics (demographic and pre-determined pandemic-related variables). Regional health authority was included to assess for potential geographic variation and resource needs. To optimise sensitivity, the higher score was used when both guardian and youth severity scores were available. Average annual income quartile was derived from the Canada Revenue Agency Individual Tax Statistics, using the first 3 digits of participants' residential postal codes. For ethnicity and gender, groups with insufficient sample size for independent analysis were combined for analysis. We evaluated the type of support services recommended by the MyHEARTSMAP-embedded algorithm, generated based on the severity and pattern of scores within each domain and participant resource access.

## Statistical analysis

We aimed to recruit 510 participants to obtain 367 completed cases, based on our previous study with a 72% follow-up rate, to provide a 95% chance of estimating a 50% prevalence in psychosocial issues to within 5%. We used descriptive statistics to summarise participant demographics and outcome measures. We compared the demographic variables for those who did and did not complete the MyHEARTSMAP assessment to test for selection bias. MyHEARTSMAP scores were summarized by domain and severity. Proportional odds models were used to measure the association between demographic variables, including regional health authorities, age, ethnicity, gender, neighbourhood household income, school status, and guardian employment, and severity of psychiatry, social, and youth health domain scores (treated as an ordinal scale). These three domains were chosen due to the nature of clinical recommendations offered as the function domain focuses on non-clinical recommendations like reducing stress. Model results are summarised as odds ratios and 95% confidence intervals. Sensitivity analysis of models with youth- and guardian-only responses was conducted [see S1 File]. Triggered recommendations per domain were summarised. Participants could trigger more than one recommendation within a domain and therefore total recommendations within a domain may exceed 100%. Duplicate identical recommendations for one individual within a domain were excluded. All analyses were conducted using R statistical software, version 4.0.3.

## Results

### Participants

A total of 675 families expressed interest in participating in this study by filling out a contact form. We obtained consent to participate from 541 families, and 78.4% (424/541) completed both the general survey and MyHEARTSMAP assessment. Of the completed assessments, 42.9% (182/424) were completed by both youth and guardian, 46.9% (199/424) were completed by guardian only, and 10.1% (43/424) were completed by youth only. The flow of participants through the study is illustrated in Fig 1. Standardised mean differences (SMD) between individuals who did and did not complete the MyHEARTSMAP assessment highlighted that individuals who did not complete were more likely to have a chronic mental health condition (SMD = 0.281, p = 0.039), or be unemployed (SMD = 0.440, p = 0.006) [S2 Table].

The median age of the youth was 10.0 years (IQR 8.0–13.0). Participants were recruited from all provincial health authorities: 28.5% (121/424) from Vancouver Coastal Health, 35.1% (149/424) from Fraser Health, 9.7% (41/424) from Interior Health, 19.3% (82/424) from Island Health, and 7.3% (31/424) from Northern Health, approximately proportional to the relative population of each region [14]. A summary of participant characteristics is provided in Table 1.

### Primary and secondary outcomes

Overall high rates of psychosocial difficulty were reported, particularly within the psychiatry and social domains, though most were mild [Fig 2]. In the psychiatry domain, 85.6% of youth were reported to experience at least mild degrees of difficulty (63.7% mild, 10.7% moderate, 11.1% severe), 70.9% in the social (64.5% mild, 5.5% moderate, 1.0% severe), 60.7% in the functional (52.8% mild, 2.1% moderate, 5.5% severe) and 45.7% in the youth health domains (40.3% mild, 0.7% moderate, 4.5% severe).

Using proportional odds models, the odds of reporting greater severity of difficulty was examined for the psychiatry, social, and youth health domains for 424 assessments [Fig 3 and S3 Table]. In checking assumptions for this model, there was no evidence for non-

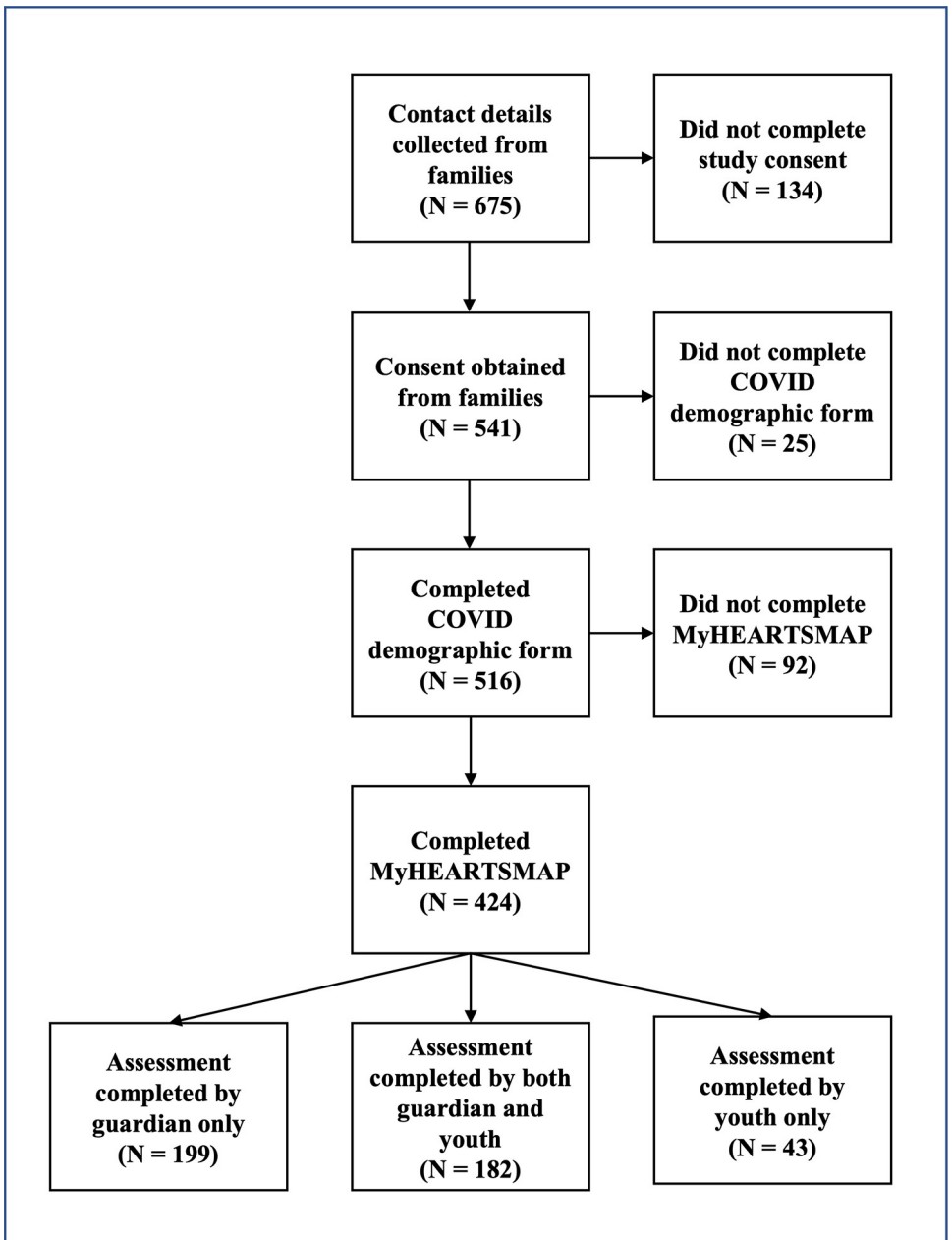

**Fig 1. Flowchart of study data collection.**

proportionality in the psychiatry (p = 0.93) or youth health domains (p = 0.27). There was some evidence of non-proportionality for the social domain (p = 0.04), however this was due to a low number of severe events (n = 4) and given our sample size we felt a multinomial model was not realistic. Older age was associated with greater severity across the psychiatry (OR = 1.29, 95% CI = 1.20, 1.39), social (OR = 1.23, 95% CI = 1.15, 1.33), and youth health (OR = 1.18, 95% CI = 1.10, 1.26) domains. Although a notably small proportion of the sample, youth with nonbinary or questioning gender identity were more likely to have greater severity in the psychiatry (OR = 4.19, 95% CI = 1.07, 16.13) and youth health (OR = 5.03, 95% CI = 1.39, 18.73) domains, while girls were more likely to report greater social difficulty

**Table 1. Participant demographic and pandemic related experience characteristics.**

| CHARACTERISTIC | N = 424 |
|---|---|
| **Age:** median years (IQR) | 10.0 (8.0, 13.0) |
| **Sex** N (%) | |
| Female | 219 (51.7) |
| Male | 204 (48.1) |
| Prefer Not to Say | 1 (0.2) |
| **Gender** N (%) | |
| Girl/Young woman | 210 (49.5) |
| Boy/Young man | 205 (48.3) |
| Non-Binary/Gender Fluid[1] | 2 (0.4) |
| Questioning/Prefer not to say[1] | 6 (1.4) |
| Unknown | 1 (0.2) |
| **Ethnicity** N (%) | |
| Asian | 27 (6.6) |
| Black or African | 4 (1.0) |
| Indigenous | 10 (2.4) |
| Hispanic | 1 (0.2) |
| Middle Eastern | 2 (0.5) |
| Multiethnic | 68 (16.6) |
| White | 297 (72.6) |
| Unknown | 15 (3.5) |
| **Guardian employment status** N (%) | |
| Employed work at home | 69 (16.3) |
| Employed work outside of home | 207 (48.8) |
| Self-employed work at home | 50 (11.8) |
| Self-employed work outside of home | 21 (5.0) |
| Unemployed | 77 (18.2) |
| **Current type of school attendance** N (%) | |
| At home (full time) | 65 (15.4) |
| In person (full time) | 200 (47.3) |
| Part time (in person) | 64 (15.1) |
| No school or formal education program | 55 (13.0) |
| Summer holiday | 39 (9.2) |
| Unknown | 1 (0.2) |
| **Neighbourhood income** median CAD (IQR) | 50,655 (44,737, 56,175) |

[1] These categories were combined for statistical analysis due to small sample size

(OR = 2.03, 95% CI = 1.32, 3.16), compared to boys. Youth who were not in school, (those not attending any formal educational program, either at-home or in-person, at a time when school would normally be in session), were more likely to report greater severity in the psychiatry (OR = 2.30, 95% CI = 1.20, 4.42) and youth health (OR = 2.10, 95% CI = 1.12, 3.94) domains, compared to youth in full-time in-person school. Youth attending school full-time at-home were more likely to report greater severity in the youth health domain (OR = 2.10, 95% CI = 1.17, 3.77). The role of regional health authority, ethnicity, neighbourhood income, or guardian employment were not significant, as confidence intervals spanned both increases and decreases in MyHEARTSMAP severity. Sensitivity analysis of models with youth- and guardian-only responses showed findings reflective of the total group [S4 & S5 Figs]. No effect

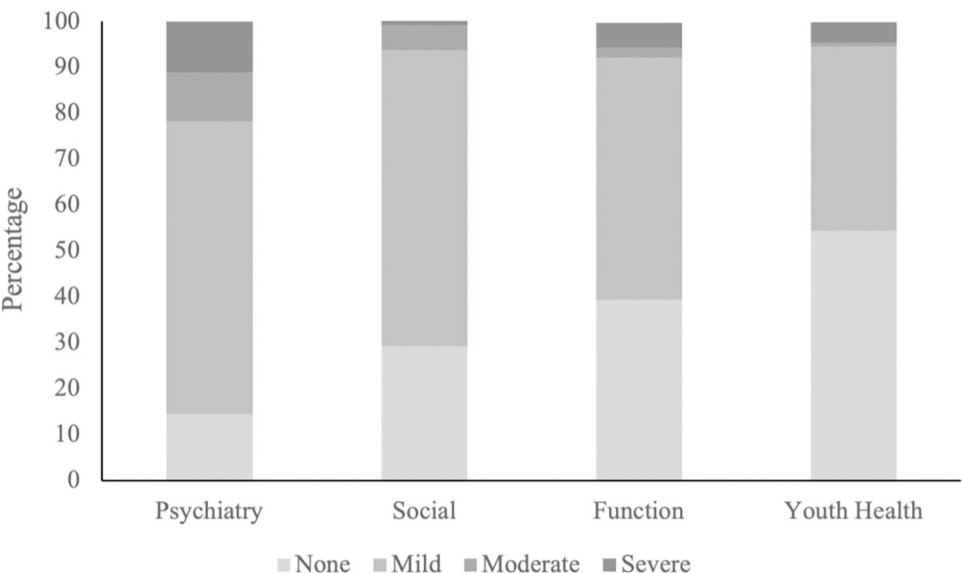

**Fig 2. Domain severity based on either guardian or youth score.**

of time was noted when included to control for the time of year and pandemic stage, so it was dropped from the model.

Recommendations to consider accessing Child and Youth Mental Health (CYMH) services to address unmet psychosocial concerns were triggered for 73.6% of youth. Acute needs for community mental health services or a severe crisis response were identified in 6.2% of youth. Specific services were recommended for 6.4% of youth from the social domain, and 11.8% of youth from the youth health domain. Table 2 outlines the recommendations generated by the MyHEARTSMAP tool for both child- and guardian-completed assessments.

## Discussion

Psychosocial concerns were observed in more than half of youth participants during the COVID-19 pandemic, though most were mild. Our primary findings are supported by research showing that pre-pandemic prevalence of mental health concerns in Canadian youth range from 18–22%, similar to 21% of respondents indicating moderate or severe psychiatric concern in our study [15]. When considering mild severity, our findings indicate psychiatric concerns in 86% of participants. Our research builds on previous work in adults, showing poor mental health outcomes during the COVID-19 pandemic for women, trans, and nonbinary people [16, 17].

Previous work using the HEARTSMAP ED assessment has shown high rates of psychiatric, social, and youth health concerns in children presenting in pediatric EDs [18]. Our findings suggest that those needs extend beyond the ED and there is a great need for community-based mental health support for children and adolescents. Our results support an urgent need for more health and prevention service provision in BC. While most previous studies focus on specific psychiatric diagnoses, the MyHEARTSMAP evaluation includes a more comprehensive screening to capture a lower threshold of psychosocial concerns and the potential for preventative early interventions.

Understanding the impact of school status on youth mental health is complex. While we found an association between increased severity of psychosocial difficulties and having no formal schooling, the nature and direction of the association could not be evaluated, as school

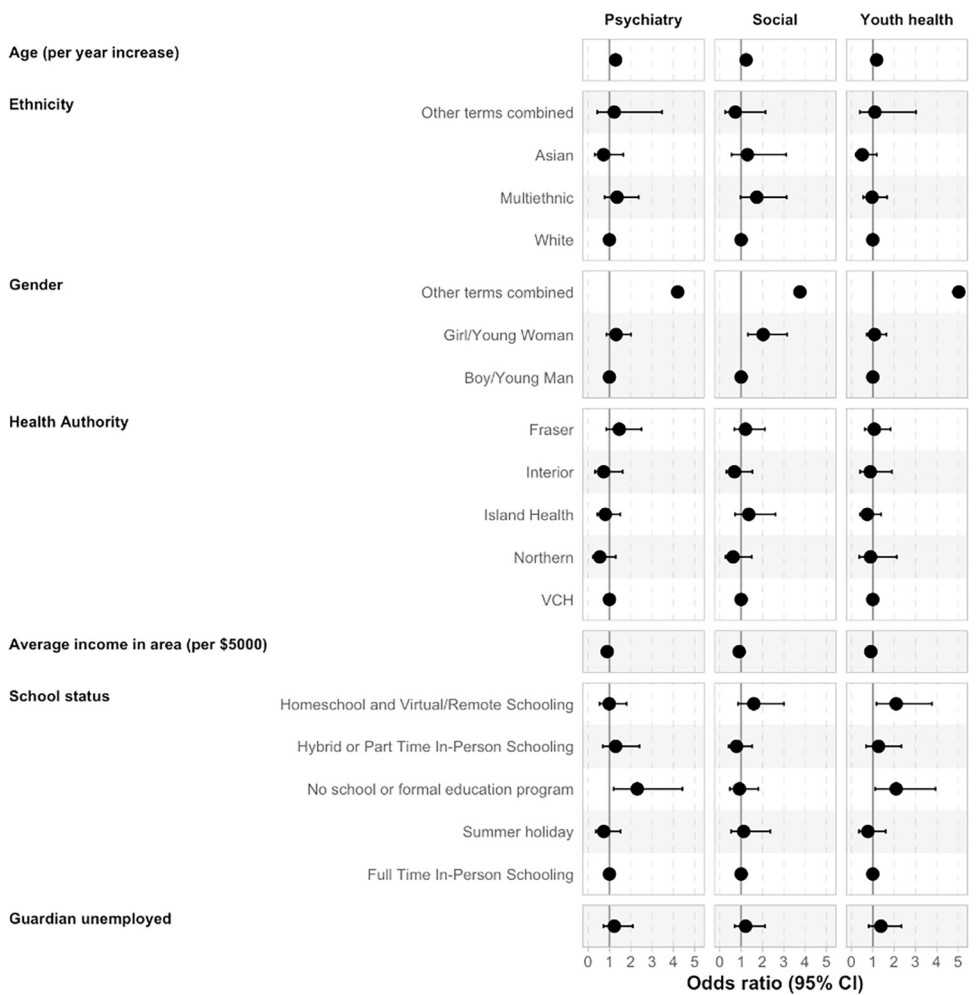

**Fig 3. Results from multivariable proportional odds model indicating the odds of increased severity score (0–7) in the psychiatry, social, and youth health domains.** Odds ratio and confidence interval values available in S1 File. No CI's for age, gender-other terms combined, and average income as intervals too large for scale of plot.

absenteeism may be due to pre-existing mental health concerns. All school closure had been reopened in BC before recruitment for this study meaning school absenteeism was related to reasons other than public health mandated closure. This challenge also applies for children who were homeschooled or in virtual or remote schooling [19, 20].

Worsening youth psychosocial status during the COVID-19 pandemic has been associated with social and physical isolation, health-related worry, conflict with parents, and difficulties with online learning [21, 22]. School closures have occurred alongside other public health measures and the ongoing stress of the COVID-19 pandemic. The interaction between these factors is unknown, thus evidence on the potential harm of school closure on youth mental health during this pandemic still needs investigation. Children of varying socio-economic backgrounds may not be equally affected by school closures and school attendance offers benefits beyond educational attainment including physical health behaviours, social services, and social support [23–25]. Further research investigating the relationship between school status and psychosocial concerns and what factors might impact difficulties with at-home learning will be essential in mitigating the impact of pandemic-related precautions on child mental wellness.

**Table 2. MyHEARTSMAP-generated mental health resource and social support recommendations for assessments completed by youths and guardians.**

| Domain | Triggered recommendation* | Total (N = 424) |
|---|---|---|
| Psychiatry | No recommendations (no identified needs)—N (%) | 61 (14.4) |
| | No new recommendations (all needs being addressed)–N (%) | 52 (12.3) |
| | Consider CYMH–N (%)[+] | 312 (73.6) |
| | Recommend CYMH–N (%) | 7 (1.7) |
| | Severe/Crisis response–N (%) | 19 (4.5) |
| Social | No recommendations (no identified needs)–N (%) | 397 (93.6) |
| | Consider MCFD or family counselling[#] - N (%) | 27 (6.4) |
| Youth Health | No recommendations (no identified needs)–N (%) | 388 (91.5) |
| | Consider primary care physician, pediatrician, Foundry, or Sex Sense–N (%) | 34 (8.0) |
| | Recommending primary care physician, pediatrician, Foundry, Sex Sense, or alcohol and substance counselling–N (%) | 16 (3.8) |

*No recommendations (no identified needs) indicate youth with low severity of concerns.

No new recommendations (all needs being addressed) indicate youth whose psychosocial concerns would trigger recommendations, but already having resources in place.

[+]CYMH–Child and Youth Mental Health services[#]MCFD—Ministry of Children and Family Development

Note that more than one recommendation might be triggered within a domain. All triggered recommendations were reported, with duplicate identical recommendations removed, so total domain recommendations exceed 100%.

## Limitations

MyHEARTSMAP is a digital, self-administered psychosocial assessment and while it shows good validity and inter-rater reliability, there are some limitations [11]. Indigenous youth in BC (9.1%) were under-represented in our sample (2%). Participation was restricted to English-speaking individuals due to the nature of the MyHEARTSMAP tool, but effort is currently underway to adapt this tool for other languages and cultures. Participants were recruited virtually, and a lack of random sampling leaves the possibility for participation bias. Our analysis of participants who did not complete the psychosocial assessment indicated selection bias, with a higher proportion of non-responders reporting chronic mental health concerns and unemployment, both potential risk factors for psychosocial concerns. However, the proportion of respondents reporting any psychosocial concerns is also consistent with other surveys during the pandemic [6, 15, 26–28]. While care was taken to recruit a geographically representative sample, other demographic factors were not the focus of this recruitment and may differ from the BC population. Recruitment occurred throughout the pandemic under varying public health restrictions. We did not observe large changes in our estimates when time was included in our models, mitigating this concern. Further research should focus on recruitment of non-binary and trans youth to further evaluate these findings.

## Conclusions

Within this community-based sample of youth in BC during the COVID-19 pandemic, we identified a high burden of psychosocial concerns. More than 50% of youth reported experiencing some degree of difficulty across psychiatry, social, or functional domains and most participants were advised to consider utilising community health services. Age, gender, and school status were associated with greater likelihood of mental health concern. Widespread mental health assessment should be available for youth in community to better identify

those dealing with psychosocial concerns and to properly allocate available mental health resources.

## Supporting information

**S1 File. Supplementary information.**
(DOCX)

**S2 File.**
(DOCX)

**S1 Fig. Screenshot of MyHEARTSMAP webpage featuring infographics for both parents and children completing the assessment.**
(TIF)

**S2 Fig. Screenshot of MyHEARTSMAP webpage featuring a sample question from the assessment.**
(TIF)

**S3 Fig. Organisation of 10 psychosocial sections into four domains used for resource recommendation by MyHEARTSMAP assessment.**
(TIF)

**S4 Fig. Results from multivariable proportional odds model indicating the odds of increased severity score (0–7) in the psychiatry, social, and youth health domains for all youth-completed assessments.**
(TIF)

**S5 Fig. Results from multivariable proportional odds model indicating the odds of increased severity score (0–7) in the psychiatry, social, and youth health domains for all guardian-completed assessments.**
(TIF)

**S1 Table. List of community organizations and groups who assisted with study recruitment through distribution of recruitment materials to their networks.**
(DOCX)

**S2 Table. Comparison of demographic variables and risk factors for individuals participated in our study and completed the MyHEARTSMAP assessment at baseline compared to those who did not.**
(DOCX)

**S3 Table. Results from multivariable proportional odds model indicating the odds of increased severity score (0–7) in the psychiatry, social, and youth health domains.**
(DOCX)

## Author Contributions

**Conceptualization:** Garth Meckler, Tyler Black, S. Evelyn Stewart, Hasina Samji, Skye Barbic, Quynh Doan.

**Data curation:** Alaina Chun, Cindy Liu.

**Formal analysis:** Jeffrey N. Bone.

**Funding acquisition:** Quynh Doan.

**Methodology:** Kaitlyn Kilyk, Jeffrey N. Bone, Garth Meckler, Quynh Doan.

**Project administration:** Alaina Chun, Cindy Liu, Quynh Doan.

**Resources:** Quynh Doan.

**Supervision:** Kaitlyn Kilyk, Garth Meckler, Tyler Black, Quynh Doan.

**Writing – original draft:** Melissa L. Woodward, Abrar Hossain.

**Writing – review & editing:** Melissa L. Woodward, Abrar Hossain, Alaina Chun, Cindy Liu, Kaitlyn Kilyk, Jeffrey N. Bone, Garth Meckler, Tyler Black, S. Evelyn Stewart, Hasina Samji, Skye Barbic, Quynh Doan.

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
