## [Decision Letter · Decision Letter 0]

22 Nov 2022

PONE-D-22-23103Evaluating the psychosocial status of BC children and youth during the COVID-19 pandemic: A MyHEARTSMAP cross-sectional study.PLOS ONE

Dear Dr. Woodward,

Thank you for submitting your manuscript to PLOS ONE. After careful consideration, we feel that it has merit but does not fully meet PLOS ONE’s publication criteria as it currently stands. Therefore, we invite you to submit a revised version of the manuscript that addresses the points raised during the review process.

We look forward to receiving your revised manuscript.

Kind regards,

Sawsan Abuhammad

Academic Editor

PLOS ONE

Journal Requirements:

Reviewers' comments:

Reviewer's Responses to Questions

**Comments to the Author**

1. Is the manuscript technically sound, and do the data support the conclusions?

Reviewer #1: Yes

2. Has the statistical analysis been performed appropriately and rigorously? 

Reviewer #1: Yes

3. Have the authors made all data underlying the findings in their manuscript fully available?

Reviewer #1: No

4. Is the manuscript presented in an intelligible fashion and written in standard English?

Reviewer #1: Yes

5. Review Comments to the Author

Reviewer #1: The topic referred to children and youth however, in line 90, authors stated that “the study aims to estimate the frequency of psychosocial and health concerns of youth and their caretakers”. I feel including caretakers in the aim has gone against the topic of the manuscript.

It will also be interesting to see some screenshots of the MyHEARTSMAP.

I feel the OR model should make use of data from youth N= 43 and youth and guardian N= 182. I see in the manuscript that the study used data from N=424. Does that mean the study included “assessment completed by guardian only in the data analysis N= 199? I suggest that authors explicitly state the sample size included in the OR model.

In table 2, the sum of youth health category 388+34+16 =438 does not add up to the total of “assessment completed by both guardian and youth” and “assessment completed by youth only” 182+43 (fig.1). Line 244 referred to 73.6% of youth which is equivalent to 312 youths. This does not add up to the total number of youths that participated in the assessment 182+42+ 225

Findings make sense. I think it would be interesting to see how the use of the platform evolved and whether the users find resources, other than nurses and health care professionals, useful in improving their mental health.

6. PLOS authors have the option to publish the peer review history of their article (what does this mean?). If published, this will include your full peer review and any attached files.

Reviewer #1: No

---

## [Author Response · Author response to Decision Letter 0]

8 Jan 2023

Dear Dr. Sawsan Abuhammad

PLOS ONE Academic Editor

Re: Revision required [PONE-D-22-23103] Evaluating the psychosocial status of BC children and youth during the COVID-19 pandemic: A MyHEARTSMAP cross-sectional study

We appreciate the thoughtful review of this manuscript and the opportunity to revise the manuscript for submission. Please see below the response to reviewers and list of revisions made.

The manuscript formatting was edited to adhere to PLOS ONE guidelines including the formatting of section headings, citation brackets, and references. The specification of receiving informed verbal consent was included in our ethics statement. No other changes were made to the reference list beyond formatting.

Comments to the Author

1. Is the manuscript technically sound, and do the data support the conclusions?

Reviewer #1: Yes

No changes were made to the manuscript

2. Has the statistical analysis been performed appropriately and rigorously?

Reviewer #1: Yes

No changes were made to the manuscript

3. Have the authors made all data underlying the findings in their manuscript fully available?

Reviewer #1: No

Data has not been made publicly available as this would violate our ethical approval due to the need for participant privacy and the potentially identifying information included in participant interviews. Data will be available for potential collaboration upon request.

4. Is the manuscript presented in an intelligible fashion and written in standard English?

Reviewer #1: Yes

No changes were made in the manuscript

5. Review Comments to the Author

Reviewer #1: The topic referred to children and youth however, in line 90, authors stated that “the study aims to estimate the frequency of psychosocial and health concerns of youth and their caretakers”. I feel including caretakers in the aim has gone against the topic of the manuscript.

Have edited this sentence to clarify that both youth and caretakers completed assessments of the mental health of the youth only. The psychosocial status of caretakers was not assessed in this study. (line 109)

It will also be interesting to see some screenshots of the MyHEARTSMAP.

A link to the MyHEARTSMAP tool and screenshots have been included in the Supplementary Information.

I feel the OR model should make use of data from youth N= 43 and youth and guardian N= 182. I see in the manuscript that the study used data from N=424. Does that mean the study included “assessment completed by guardian only in the data analysis N= 199? I suggest that authors explicitly state the sample size included in the OR model.

I believe this mistake was due to our previous lack of clarity around caretaker assessments. The OR model included all 424 assessments. When assessments of youth mental health were completed by both the youth and their guardian, the higher severity rating was included to increase sensitivity. The sample size of the OR model has been added. (line 290)

In table 2, the sum of youth health category 388+34+16 =438 does not add up to the total of “assessment completed by both guardian and youth” and “assessment completed by youth only” 182+43 (fig.1). Line 244 referred to 73.6% of youth which is equivalent to 312 youths. This does not add up to the total number of youths that participated in the assessment 182+42+ 225

As noted in the footnote for this table, more than one recommendation may be triggered for different questions within a domain and therefore the total number of recommendations within a domain may exceed 100%. (line 346) This note was also added to the methods to promote clarity. (line 234)

Findings make sense. I think it would be interesting to see how the use of the platform evolved and whether the users find resources, other than nurses and health care professionals, useful in improving their mental health.

We are currently analysing the data from the three-month follow-up study and preparing that work for publication so we are excited to hear that this is a matter of interest for the reviewer.

Thank you for your time and thought for this review in improving our manuscript.

Sincerely,

Melissa Woodward

Postdoctoral Fellow

---

## [Editor Report · Decision Letter 1]

16 Jan 2023

Evaluating the psychosocial status of BC children and youth during the COVID-19 pandemic: A MyHEARTSMAP cross-sectional study.

PONE-D-22-23103R1

Dear Dr. Woodward,

We’re pleased to inform you that your manuscript has been judged scientifically suitable for publication and will be formally accepted for publication once it meets all outstanding technical requirements.

Kind regards,

Sawsan Abuhammad

Academic Editor

PLOS ONE
---

## [Editor Report · Acceptance letter]

20 Mar 2023

PONE-D-22-23103R1 

Evaluating the psychosocial status of BC children and youth during the COVID-19 pandemic: A MyHEARTSMAP cross-sectional study. 

Dear Dr. Woodward:

I'm pleased to inform you that your manuscript has been deemed suitable for publication in PLOS ONE. Congratulations! Your manuscript is now with our production department. 

Kind regards, 

on behalf of

Dr. Sawsan Abuhammad 

Academic Editor

PLOS ONE